# Smoothed Embeddings for Certified Few-Shot Learning

**Mikhail Pautov**
Skolkovo Institute of Science and Technology

**Olesya Kuznetsova**[*]
Skolkovo Institute of Science and Technology

**Nurislam Tursynbek**
The University of North Carolina at Chapel Hill

**Aleksandr Petiushko**
Moscow State University, Nuro, Inc.

**Ivan Oseledets**
Skolkovo Institute of Science and Technology, AIRI

## Abstract

Randomized smoothing is considered to be the state-of-the-art provable defense against adversarial perturbations. However, it heavily exploits the fact that classifiers map input objects to class probabilities and do not focus on the ones that learn a metric space in which classification is performed by computing distances to embeddings of class prototypes. In this work, we extend randomized smoothing to few-shot learning models that map inputs to normalized embeddings. We provide analysis of the Lipschitz continuity of such models and derive a robustness certificate against $\ell_2$-bounded perturbations that may be useful in few-shot learning scenarios. Our theoretical results are confirmed by experiments on different datasets.

## 1 Introduction

Few-shot learning is a setting in which a classification model is evaluated on the classes not seen during the training phase. Nowadays quite a few few-shot learning approaches based on neural networks are known. Unfortunately, neural network-based classifiers are intimately vulnerable to adversarial perturbations [41, 10] – accurately crafted small modifications of the input that may significantly alter the model's prediction. This vulnerability poses a restriction on the deployment of such approaches in safety-critical scenarios, so the research interest in the field of attacks on neural networks has been great in recent years.

Several works studied this phenomena in different practical applications – image classification [2, 34, 33, 40], object detection [17, 26, 48, 50], face recognition [18, 5, 54], semantic segmentation [7, 12, 50]. These studies show that it is easy to force a model to behave in the desired way by applying an imperceptible change to its input. As a result, defenses, both empirical [4, 55, 14] and provable [51, 21, 3, 47, 52, 15, 46, 35], were proposed recently. Although empirical ones can be (and often are) broken by more powerful attacks, the provable ones are of a big interest since they make it possible to provide *guarantees* of the correctness of the work of a model under certain assumptions, and, thus, possibly broaden the scope of tasks which may be trusted to the neural networks.

Randomized smoothing [21, 3, 25] is the state-of-the-art approach used for constructing classification models provably robust against small-norm additive adversarial perturbations. This approach is scalable to large datasets and can be applied to any classifier since it does not use any assumptions

---

[*]The code for this paper is available at `https://github.com/koava36/certrob-fewshot`.

36th Conference on Neural Information Processing Systems (NeurIPS 2022).

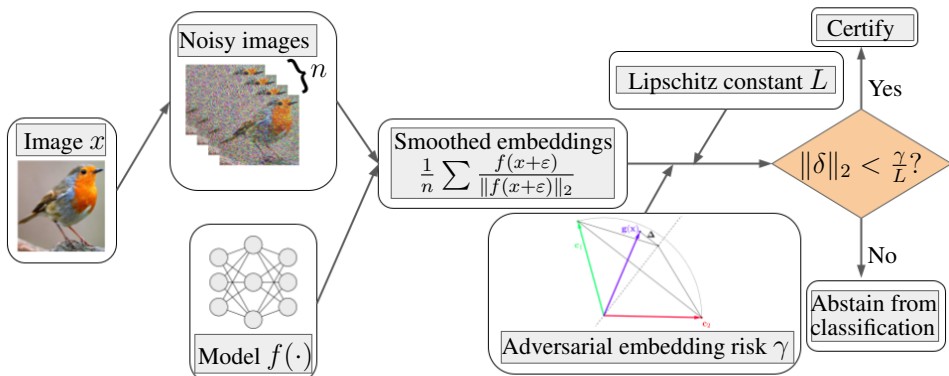

Figure 1: Illustration of certification pipeline for a single image $x$. Given a model $f(\cdot)$ and $n$ realisations of zero-mean Gaussian noise $\varepsilon_1, \ldots, \varepsilon_n \sim \mathcal{N}(0, \sigma^2 I)$, an estimate $\hat{g}(x) = \frac{1}{n}\sum_{i=1}^{n} f(x + \varepsilon_i)$ of $g(x) = \mathbb{E}_{\varepsilon \sim \mathcal{N}(0,\sigma^2 I)} f(x + \varepsilon)$ is computed. Note that $g(x)$ is $L$-Lipschitz with $L = \sqrt{2/\pi\sigma^2}$, according to the Theorem 1. The number of samples $n$ is increased until adversarial embedding risk $\gamma$ from the Theorem 2 is computed with Algorithms 1-2 and certified radius $r = \frac{\gamma}{L}$ is determined. The model $g(\cdot)$ is treated as certified at $x$ for all additive perturbations $\delta : \|\delta\|_2 < r$.

about the model's architecture. Generally, the idea is the following. Suppose, we are given a base neural network classier $f : \mathbb{R}^D \to [0, 1]^K$ that maps an input image $x$ to a fixed number of $K$ class probabilities. Its smoothed version with the standard Gaussian distribution is:

$$g(x) = \mathbb{E}_{\varepsilon \sim \mathcal{N}(0,\Sigma)} f(x + \varepsilon). \tag{1}$$

As shown in [3], the new (smoothed) classifier is provably robust at $x$ to $\ell_2$-bounded perturbations if the base classifier $f$ is confident enough at $x$. However, the proof of certification heavily exploits the fact that classifiers are restricted to mapping an input to a fixed number of class probabilities. Thus, directly applying randomized smoothing to classifiers in metric space, such as in few-shot learning, is a challenging task.

There are several works that aim at improving the robustness for few-shot classification [20, 9, 53, 28]. However, the focus in such works is either on the improvement of empirical robustness or probabilistic guarantees of certified robustness; none of them provide theoretical guarantees on the worst-case model behavior.

In this work, we fill this gap, by generalizing and theoretically justifying the idea of randomized smoothing to few-shot learning. In this scenario, provable certification needs to be obtained not in the space of output class probabilities, but in the space of descriptive embeddings. This work is the first, to our knowledge, where the theoretical robustness guarantees for a few-shot scenario are provided.

**Our contributions are summarized as follows:**

- We provide the first theoretical robustness guarantee for few-shot learning classification.

- Analysis of Lipschitz continuity of such models and providing the robustness certificates against $\ell_2-$bounded perturbations for few-shot learning scenarios.

- We propose to estimate confidence intervals not for distances between the approximation of smoothed embedding and class prototype but for the dot product of vectors which has expectation equal to the distance between actual smoothed embedding and class prototype.

## 2 Problem statement

### 2.1 Notation

We consider a few-shot classification problem where we are given a set of labeled objects $(x_1, y_1), \ldots, (x_m, y_m)$ where $x_i \in \mathbb{R}^D$ and $y_i \in \{1, \ldots, K\}$ are corresponding labels. We follow the notation from [39] and denote $S_k$ as the set of objects of class $k$.

## 2.2 Few-shot learning classification

Suppose we have a function $f : \mathbb{R}^D \to \mathbb{R}^d$ that maps input objects to the space of normalized embeddings. Then, $d-$dimensional prototypes of classes are computed as follows (expression is given for the prototype of class $k$):

$$c_k = \frac{1}{|S_k|} \sum_{x \in S_k} f(x). \tag{2}$$

In order to classify a sample, one should compute the distances between its embedding and class prototypes – a sample is assigned to the class with the closest prototype. Namely, given a distance function $\rho : \mathbb{R}^d \times \mathbb{R}^d \to [0, +\infty)$, the class $c$ of the sample $x$ is computed as below:

$$c = \underset{k \in \{1,\dots,K\}}{\arg\min} \rho\left(f(x), c_k\right). \tag{3}$$

Given an embedding function $f$, our goal is to construct a classifier $g$ provably robust to additive perturbations $\Delta$ of a small norm. In other words, we want to find a norm threshold $t$ such that equality

$$\underset{k \in \{1,\dots,K\}}{\arg\min} \rho\left(g(x), c_k\right) = \underset{k \in \{1,\dots,K\}}{\arg\min} \rho\left(g(x+\delta), c_k\right), \tag{4}$$

will be satisfied for all $\delta : \|\delta\|_2 \le t$.

In this paper, the solution to a problem of constructing a classifier that satisfies the condition in (4) is approached by extending the analysis of the robustness of smoothed classifiers described in (1) to the case of vector functions. The choice of the distance metric in (4) is motivated by an analysis of Lipschitz-continuity given in the next section.

## 3 Randomized smoothing

### 3.1 Background

Randomized smoothing [21, 3] is described as a technique of convolving a base classifier $f$ with an isotropic Gaussian noise such that the new classifier $g(x)$ returns the most probable prediction of $f$ of a random variable $\xi \sim \mathcal{N}(x, \sigma^2 I)$, where the choice of Gaussian distribution is motivated by the restriction on $g$ to be robust against additive perturbations of the bounded norm. In this case, given a classifier $f : \mathbb{R}^D \to [0, 1]$ and smoothing distribution $\mathcal{N}(0, \sigma^2 I)$, the classifier $g$ looks as follows:

$$g(x) = \frac{1}{(2\pi\sigma^2)^{\frac{n}{2}}} \int_{\mathbb{R}^D} f(x+\varepsilon) \exp\left(-\frac{\|\varepsilon\|_2^2}{2\sigma^2}\right) d\varepsilon. \tag{5}$$

One can show by Stein's Lemma that given the fact that the function $f$ in (5) is bounded (namely, $\forall x \in D(f), |f(x)| \le 1$), then the function $g$ is $L-$Lipschitz:

$$\forall x, x' \in D(g), \ \|g(x) - g(x')\|_2 \le L\|x - x'\|_2, \tag{6}$$

with $L = \sqrt{\frac{2}{\pi\sigma^2}}$, what immediately produce theoretical robustness guarantee on $g$.

Although this approach is simple and effective, it has a serious drawback: in practice, it is impossible to compute the expectation in (5) exactly and, thus, impossible to compute the prediction of the smoothed function $g$ at any point. Instead the integral is computed with the use of Monte-Carlo approximation with $n$ samples to obtain the prediction with an arbitrary level of confidence. Notably, to achieve an appropriate accuracy of the Monte-Carlo approximation, the number of samples $n$ should be large enough that may dramatically affect inference time.

In this work, we generalize the analysis of Lipschitz-continuity to the case of vector functions $g : \mathbb{R}^D \to \mathbb{R}^d$ and provide robustness guarantees for classification performed in the space of embeddings. The certification pipeline is illustrated in Figure 1.

## 3.2 Randomized smoothing for vector functions

**Lipschitz-continuity of vector function.** In the work of [38], the robustness guarantee from [3] is proved by estimating the Lipschitz constant of a smoothed classifier. Unfortunately, a straightforward generalization of this approach to the case of vector functions leads to the estimation of the expectation of the norm of a multivariate Gaussian which is known to depend on the number of dimensions of the space. Instead, we show that a simple adjustment to this technique may be done such that the estimate of the Lipschitz constant is the same as for the function in (5). Our results are formulated in the theorems below proofs of which are moved to the Appendix in order not to distract the reader.

**Theorem 1.** *(Lipschitz-continuity of smoothed vector function) Suppose that $f : \mathbb{R}^D \to \mathbb{R}^d$ is a deterministic function and $g(x) = \mathbb{E}_{\varepsilon \sim \mathcal{N}(0,\sigma^2 I)} f(x + \varepsilon)$ is continuously differentiable for all $x$. If for all $x$, $\|f(x)\|_2 = 1$, then $g(x)$ is $L-$Lipschitz in $l_2-$norm with $L = \sqrt{\frac{2}{\pi \sigma^2}}$.*

**Remark 1.** *We perform the analysis of Lipschitz-continuity in the Theorem 1 in $l_2-$norm, so the distance metric in (4) is $l_2-$distance. We do not consider other norms in this paper.*

**Robust classification in the space of embeddings.** To provide certification for a classification in the space of embeddings, one should estimate the maximum deviation of the classified embedding that does not change the closest class prototype. In the Theorem 2, we show how this deviation is connected with the mutual arrangement of embedding and class prototype.

**Theorem 2.** *(Adversarial embedding risk) Given an input image $x \in \mathbb{R}^D$ and the embedding $g : \mathbb{R}^D \to \mathbb{R}^d$ the closest point on to decision boundary in the embedding space (see Figure 2) is located at a distance (defined as adversarial embedding risk):*

$$\gamma = \|\Delta\|_2 = \frac{\|c_2 - g(x)\|_2^2 - \|c_1 - g(x)\|_2^2}{2\|c_2 - c_1\|_2^2}, \tag{7}$$

*where $c_1 \in \mathbb{R}^d$ and $c_2 \in \mathbb{R}^d$ are the two closest prototypes. The value of $\gamma$ is the distance between classifying embedding and the decision boundary between classes represented by $c_1$ and $c_2$. Note that this is the minimum $l_2-$distortion in the embedding space required to change the prediction of $g$.*

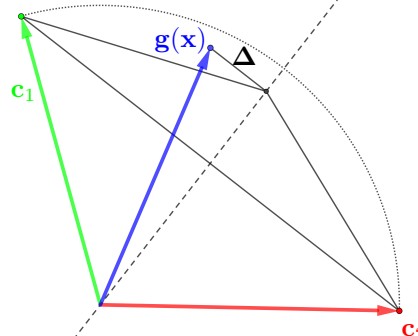

Figure 2: Illustration of the Theorem 2, one-shot case. The direction of adversarial risk in the space of embeddings is always parallel to the vector $c_1 - c_2$. This is also true for the case of $\|c_1\|_2 \neq \|c_2\|_2$.

Two theorems combined result in an $l_2-$robustness guarantee for few-shot classification as:

**Theorem 3.** *(Robustness guarantee) $L_2$-robustness guarantee $r$ for an input image $x$ in the $n-$dimensional input metric space under classification by a classifier $g$ from the Theorem 1 is $r = \frac{\gamma}{L}$, where $L$ is the Lipschitz constant from the Theorem 1 and $\gamma$ is the adversarial risk from the Theorem 2. The value of $r$ is the certified radius of $g$ at $x$, or, in other words, minimum $l_2-$distortion in the input space required to change the prediction of $g$. The proof of this fact straightforwardly follows from the definition (6) and results from Theorems 1-2.*

## 4 Certification protocol

In this section, we describe the numerical implementation of our approach and estimate its failure probability.

### 4.1 Estimation of prediction of smoothed classifier

As mentioned previously, the procedure of few-shot classification is performed by assigning an object $x$ to the closest class prototype. Unfortunately, given the smoothed function $g$ in the form from the Theorem 1 and class prototype $c_k$ from (2), it is impossible to compute the value $\rho(g(x), c_k)$ explicitly as well as to determine the closest prototype, since it is in general unknown how does $g(x)$ look like.

We propose both to estimate the closest prototype for classification and to estimate the distance to the closest decision boundary $\gamma$ from Theorem 2 as the largest class-preserving perturbation in the embedding space by computing two-sided confidence intervals for random variables

$$\xi_1 = \|\hat{g}(x) - c_1\|_2^2, \ldots, \xi_K = \|\hat{g}(x) - c_K\|_2^2, \tag{8}$$

where $\hat{g}(x) = \frac{1}{n} \sum_{i=1}^n f(x + \varepsilon_i)$ is the estimation of $g(x)$ computed as empirical mean by $n$ samples of noise, and one-sided confidence interval for $\gamma$ from the Theorem 2, respectively. Pseudo-code for both procedures is presented in Algorithms 1-2.

---

**Algorithm 1** Closest prototype computation algorithm.

**GIVEN:** base classifier $f$, noise $\sigma$, object $x$, number of samples $n$ for $\hat{g}(x)$, class prototypes $\{c_i\}_{i=1}^K$, maximum number of samples $T$ and confidence level $\alpha$,
**RETURNS:** index $A$ of the closest prototype.

---

Function CLOSEST($f, \sigma, x, n, \{c_i\}_{i=1}^K, T, \alpha$)
  $fs \leftarrow \emptyset$
  $i \leftarrow 0$
  $n_0 = n$
  **while** $n \leq T$ **do**
    $\varepsilon_1, \ldots, \varepsilon_n \sim \mathcal{N}(0, \sigma^2 I)$
    $\hat{g}(x) \leftarrow \frac{1}{n} \sum_{i=1}^n f(x + \varepsilon_i)$
    $fs \leftarrow fs \cup \{\hat{g}(x)\}$
    **for all** $i \in [1, \ldots, K]$ **do**
      $C = c_i$
      $dsToC \leftarrow \rho(fs, C)$
      $(l_C, u_C) \leftarrow \texttt{TwoSidedConfInt}(dsToC, \alpha)$
      $l_i \leftarrow l_C, u_i \leftarrow u_C$
    **end for**
    $A \leftarrow \arg\min \{l_i\}_{i=1}^K$
    **if** $u_A < \min(\{l_i\}_{i=1, i \neq A}^K)$ **then**
      Return A, fs
    **else**
      $n \leftarrow n + n_0$ {Increase number of samples used for computing an approximation $\hat{g}(x)$ until the number of observations is large enough to determine two leftmost intervals or until $n = T$}
    **end if**
  **end while**
EndFunction

---

The Algorithm 1 describes an inference procedure for the smoothed classifier from the Theorem 1; the Algorithm 2 uses Algorithm 1 and, given input parameters, estimates an adversarial risk from the Theorem 2 – it determines two closest to the smoothed embedding $g(x)$ prototypes $A$ and $B$ and produces the lower confidence bound for the distance between $g(x)$ and the decision boundary between $A$ and $B$. Combined with analysis from the Theorem 1, it provides the certified radius for a sample – the smallest value of $l_2$−norm of perturbation in the input space required to change the prediction of the smoothed classifier. In the next subsection, we discuss in detail the procedure of computing confidence intervals in Algorithms 1-2.

## 4.2 Applicability of algorithms

The computations of smoothed function and distances to class prototypes and decision boundary in Algorithms 1-2 are based on the estimations of corresponding random variables, thus, it is necessary to analyze the applicability of the algorithms. In this section, we propose a way to compute confidence intervals for squares of the distances between the estimates of embeddings and class prototypes.

**Computation of confidence intervals for the squares of distances.** Recall that one way to estimate the value of a parameter of a random variable is to compute a confidence interval for the corresponding statistic. In this work, we construct intervals by applying well-known Hoeffding inequality [13] in the form

$$\mathbb{P}\left(|\overline{X} - \mathbb{E}(\overline{X})| \geq t\right) \leq 2\exp\left(-\frac{2t^2 n^2}{\sum_{i=1}^{n}(b_i - a_i)^2}\right), \tag{9}$$

where $\overline{X}$ and $\mathbb{E}(\overline{X})$ are sample mean and population mean of random variable $X$, respectively, $n$ is the number of samples and numbers $a_i, b_i$ are such that $\mathbb{P}(X_i \in (a_i, b_i)) = 1$.

---

**Algorithm 2** Adversarial embedding risk computation algorithm.

**GIVEN:** base classifier $f$, noise $\sigma$, object $x$, number of samples $n$ for $\hat{g}(x)$, class prototypes $\{c_i\}_{i=1}^{K}$, maximum number of samples $T$ for $\hat{g}(x)$ and confidence level $\alpha$

**RETURNS:** lower bound $\Gamma$ for the adversarial risk $\gamma$.

---
Function EMBEDDING-RISK$(f, \sigma, x, n, \{c_i\}_{i=1}^{K}, T, \alpha)$
$A,\ fs_A \leftarrow$ CLOSEST$(f, \sigma, x, n, \{c_i\}_{i=1}^{K}, T, \alpha)$
$B,\ fs_B \leftarrow$ CLOSEST$(f, \sigma, x, n, \{c_i\}_{i=1, i\neq A}^{K}, T, \alpha)$
$fs \leftarrow fs_A \cup fs_B$
$\gamma s \leftarrow \emptyset$
**for all** $f \in fs$ **do**
$\quad \gamma = \frac{\|c_B - g\|^2 - \|c_A - g\|^2}{2\|c_B - c_A\|^2}$
$\quad \gamma s \leftarrow \gamma s \cup \{\gamma\}$
**end for**
$\Gamma \leftarrow$ `LowerConfBound`$(\gamma s, \alpha)$
Return $\Gamma$
EndFunction

---

However, a confidence interval for the distance $\|\hat{g}(x) - c_k\|_2$ with a certain confidence covers an *expectation of distance* $\mathbb{E}(\|\hat{g}(x) - c_k\|_2)$, not the *distance for expectation* $\|\mathbb{E}(\hat{g}(x) - c_k)\|_2 = \|g(x) - c_k\|_2$.

To solve this problem, we propose to compute confidence intervals for the dot product of vectors. Namely, given a quantity $\hat{\xi}_{x,k} = \langle g(x) - c_k, g(x) - c_k \rangle$, we sample its unbiased estimate with at maximum $2n$ samples of noise (here we have to mention that the number of samples $n$ from Algorithm 1 actually doubles since we need a pair of estimates $\hat{g}(x)$ of smoothed embeddings):

$$\hat{\xi}_{x,k} = \left\langle \frac{1}{n}\sum_{i=1}^{n} f(x + \varepsilon_i) - c_k, \frac{1}{n}\sum_{j=n+1}^{2n} f(x + \varepsilon_j) - c_k \right\rangle \tag{10}$$

and compute confidence interval $(l_{x,k}, u_{x,k})$ such that given

$$\frac{\alpha}{3} = 2\exp\left(-\frac{2t^2 n^2}{\sum_{i=1}^{n}(b_i - a_i)^2}\right), \tag{11}$$

the population mean $\mathbb{E}(\hat{\xi}_{x,k})$ is most probably located within it, or, equivalently, $\mathbb{P}\left(l_{x,k} \leq \mathbb{E}(\hat{\xi}_{x,k}) \leq u_{x,k}\right) \geq 1 - \alpha$. We have to mention that there are three confidence intervals for three terms (one with quadratic number of samples and two with the linear numbers of samples) in the expression (10), that is why there is fraction $\frac{1}{3}$ in the equation (11).

Also note that the population mean $\mathbb{E}(\hat{\xi}_{x,k})$ is exactly $\|g(x) - c_k\|_2^2$, since

$$\mathbb{E}(\hat{\xi}_{x,k}) = \mathbb{E}\left\langle \frac{1}{n}\sum_{i=1}^{n} f(x + \varepsilon_i) - c_k, \frac{1}{n}\sum_{j=n+1}^{2n} f(x + \varepsilon_j) - c_k \right\rangle = \|g(x) - c_k\|_2^2 \tag{12}$$

since $f(x + \varepsilon_i)$ and $f(x + \varepsilon_j)$ are independent random variables for $i \neq j$. Finally, note that the confidence interval $(l_{x,k}, u_{x,k})$ for the quantity $\|g(x) - c_k\|_2^2$ implies confidence interval

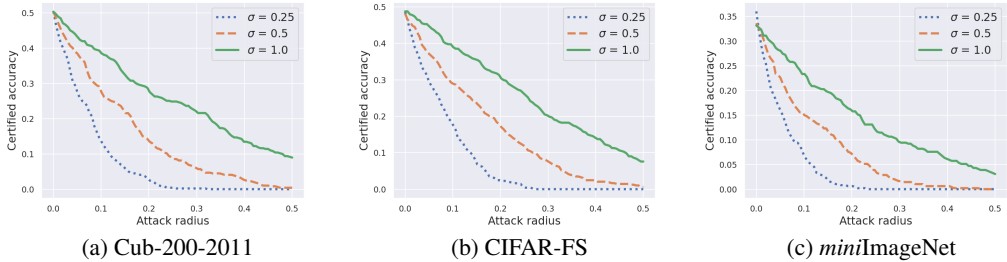

(a) Cub-200-2011        (b) CIFAR-FS        (c) *mini*ImageNet

Figure 3: Dependency of certified accuracy on attack radius $\varepsilon$ for different $\sigma$, 1-shot case, $n = 1000$.

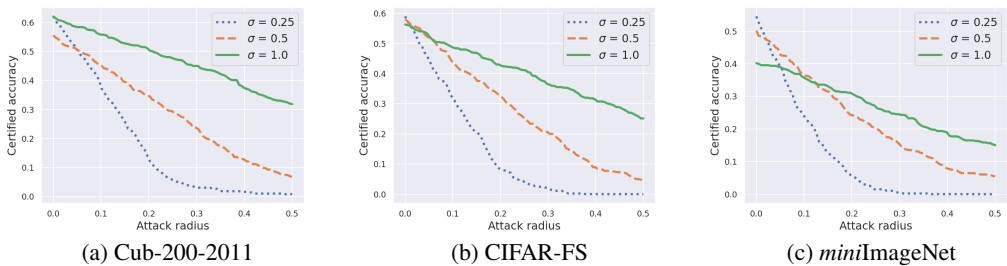

(a) Cub-200-2011        (b) CIFAR-FS        (c) *mini*ImageNet

Figure 4: Dependency of certified accuracy on attack radius $\varepsilon$ for different $\sigma$, 5-shot case, $n = 1000$.

$$\left(\sqrt{l_{x,k}}, \sqrt{u_{x,k}}\right) \tag{13}$$

for the quantity $\|g(x) - c_k\|_2$. Thus, the procedures `TwoSidedConfInt` and `LowerConfBound` from algorithms return an interval from (13) and its left bound for the random variable representing corresponding distance, respectively.

## 5 Experiments

### 5.1 Datasets

For the experimental evaluation of our approach we use several well-known datasets for few-shot learning classification. *Cub-200-2011* [45] is a dataset with $11,788$ images of 200 bird species, where 5864 images of 100 species are in the train subset and 5924 images of other 100 species are in the test subset. It is notable that a lot of species presented in dataset have degree of visual similarity, making classification of ones a challenging task even for humans. *mini*ImageNet [44] is a subset of images from *ILSVRC 2015* [37] dataset with 64 images categories in train subset, 16 categories in validation subset and 20 categories in test subset with 600 images of size $84 \times 84$ in each category. *CIFAR FS* [1] is a subset of *CIFAR 100* [19] dataset which was generated in the same way as *mini*ImageNet and contains 37800 images of 64 categories in the train set and 11400 images of 20 categories in the test set. Experimental setup for all the datasets is presented in the next section.

### 5.2 Experimental settings and computation cost

Following [3], we compute approximate certified test set accuracy to estimate the performance of the smoothed model prediction with the Algorithm 1 and embedding risk computation with the Algorithm 2. The baseline model we used for experiments is a prototypical network introduced in [39] with ConvNet-4 backbone. Compared to the original architecture, an additional fully-connected layer was added in the tail of the network to map embeddings to 512-dimensional vector space. The model was trained to solve 1-shot and 5-shot classification tasks on each dataset, with 5-way classification on each iteration.

**Parameters of expeiments.** For data augmentation, we applied Gaussian noise with zero mean, unit variance and probability 0.3 of augmentation. Each dataset was certified on a subsample of 500 images with default parameters for the Algorithm 1: number of samples $n = 1000$, confidence level $\alpha = 0.001$ and variance $\sigma = 1.0$, unless stated otherwise. For our settings, it may be shown from simple geometry that values $(a_i, b_i)$ from (9) are such that $b_i - a_i \leq 4$ so we use $b_i - a_i = 4$. The maximum number of samples $T$ in the Algorithm 1 is set to be $T = 5 \times 10^5$.

**Computation cost.** In the table below, we report the computation time of the certification procedure per image on Tesla V100 GPU for *Cub-200-2011* dataset. Standard deviation in seconds appears to be significant because the number of main loop iterations required to separate the two leftmost confidence intervals varies from image to image in the test set.

Table 1: Computation time per image of implementation of the Algorithm 2, Cub-200-2011.

| n | $10^3$ | $10^4$ | $10^5$ |
|---|---|---|---|
| t, sec | $0.044 \pm 0.030$ | $0.509 \pm 0.403$ | $4.744 \pm 2.730$ |

## 5.3 Results of experiments

In this section, we report the results of our experiments. In our evaluation protocol, we compute approximate certified test set accuracy, $CA$. Given a sample $x$, a smoothed classifier $g(\cdot)$ from the Algorithm 1 with an assigned classification rule $h(x) = \arg\min_{i \in \{1,...,K\}} \|g(x) - c_k\|_2$, threshold value $\varepsilon$ for $l_2-$norm of additive perturbation and the robustness guarantee $r = r(x)$ from the Algorithm 3, we compute $CA$ on test set $S$ as follows:

$$CA(S, \varepsilon) = \frac{|(x,y) \in S : r(x) > \varepsilon \ \& \ h(x) = y|}{|S|}. \tag{14}$$

In other words, we treat the model $g(\cdot)$ as certified at point $x$ under perturbation of norm $\varepsilon$ if $x$ is correctly classified by $g(\cdot)$ (what means that the procedure of classification described in the Algorithm 1 does not abstain from classification of $x$) and $g(\cdot)$ has the value of certified radius $r(x) > \varepsilon$.

**Visualization of results.** The figures 3-4 represent dependencies of certified accuracy on the value of norm of additive perturbation for different learning settings (1-shot and 5-shot learning). The value of the attack radius corresponds to the threshold $\varepsilon$ from (14). For *Cub-200-2011* dataset we provide a dependency of certified accuracy for different sample size $n$ for the Algorithm 1 (in the Figure 5).

# 6 Limitations

In this section, we provide failure probability of Algorithms 1-2, discuss abstains from classification in the Algorithm 1 and speculate on the application of our method in other few-shot scenarios.

## 6.1 Estimation of errors of algorithms

Note that the value of $\alpha$ from (11) is the probability of the value of $\rho_{x,k} = \|g(x) - c_k\|_2$ not to belong to the corresponding interval of the form from (13). Given a sample $x$, the procedure in the Algorithm 1 returns two closest prototypes to the $g(x)$. To determine two leftmost confidence intervals, all the distances $\rho_{x,k}$ have to be located within corresponding intervals, thus, according to the independence of computing these two intervals, the error probability for the Algorithm 1 is $q_1 = K\alpha$, where $K$ is the number of classes. Similarly, the procedure in the Algorithm 2 outputs the lower bound for the adversarial risk with coverage at least $1 - \alpha$ and depend on the output of the Algorithm 1 inside, and, thus, has error probability $q_2 = 1 - (1 - \alpha)(1 - K\alpha) = \alpha + K\alpha - K\alpha^2$ that corresponds to returning an overestimated lower bound for the adversarial risk from the Theorem 2.

## 6.2 Abstains from classification and extension to other few-shot approaches

It is crucial to note that the procedure in the Algorithm 1 may require a lot of samples to distinguish two leftmost confidence intervals and sometimes does not finish before reaching threshold

value $T$ for sample size. As a result, there may be input objects at which the smoothed classifier can be neither evaluated nor certified. In this subsection, we report the fraction of objects in which the Algorithm 1 abstains from determining the closest class prototype (see Tables 2-3).

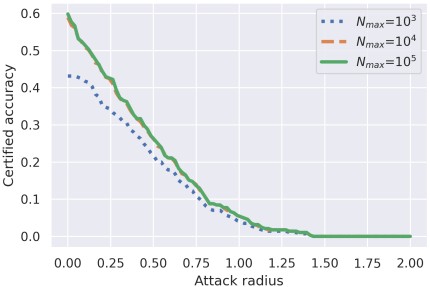

Figure 5: Dependency of certified accuracy on attack radius $\varepsilon$ for different number of samples $n$ of the Algorithm 1, *CIFAR-FS* dataset, 5-shot case. It is notable that a relatively small number of samples may be used to achieve a satisfactory level of certified accuracy.

Table 2: Percentage of non-certified objects in test subset, 1-shot case.

|  | $\alpha = 10^{-2}$ | $\alpha = 10^{-3}$ | $\alpha = 10^{-4}$ |
|---|---|---|---|
| Cub-200-2011 | 1.6% | 1.6% | 1.6% |
| CIFAR-FS | 2.2% | 2.2% | 2.4% |
| *mini*ImageNet | 1.9% | 2.2% | 2.4% |

Table 3: Percentage of non-certified objects in test subset, 5-shot case.

|  | $\alpha = 10^{-2}$ | $\alpha = 10^{-3}$ | $\alpha = 10^{-4}$ |
|---|---|---|---|
| Cub-200-2011 | 1.2% | 1.2% | 1.4% |
| CIFAR-FS | 3.0% | 3.4% | 3.8% |
| *mini*ImageNet | 2.9% | 2.9% | 3.0% |

Since our method is based on randomized smoothing, among the approaches presented in [43] it is applicable for matching networks and to some extent to MAML networks. In the case of matching networks, the new sample is labeled as weighted cosine distance to support samples, so it is easy to transfer guarantees for $l_2-$distance to the ones for cosine distance in case of normalized embeddings. For MAML, smoothing may be applied to the embedding function as well, but theoretical derivations of certificates are required.

## 7 Related work

Breaking neural networks with adversarial attacks and empirical defending from them have a long history of cat-and-mouse game. Namely, for a particular proposed defense against existing adversarial perturbations, a new more aggressive attack is found. This motivated researchers to find defenses that are mathematically provable and certifiably robust to different kinds of input manipulations. Several works proposed exactly verified neural networks based on Satisfiability Modulo Theories solvers [16, 6], or mixed-integer linear programming [30, 8]. These methods are found to be computationally inefficient, although they guarantee to find adversarial examples, in the case they exist. Another line of works use more relaxed certification [47, 11, 36]. Although these methods aim to guarantee that an adversary does not exist in a certain region around a given input, they suffer from scalability to big networks and large datasets. The only scalable to large datasets provable defense against adversarial perturbations is *randomized smoothing*. Initially, it was found as an empirical defense to mitigate adversarial effects in neural networks [29, 49]. Later several works showed its mathematical proof of certified robustness [21, 25, 3, 38]. Lecuyer et al [21] first provided proof of certificates against adversarial examples using differential privacy. Later, Cohen et al [3] provided the tightest bound using Neyman-Pearson lemma. Interestingly, alternative proof using Lipschitz continuity was found [38]. The scalability and simplicity of randomized smoothing attracted significant attention, and it was extended beyond $l_2-$perturbations [22, 42, 27, 24, 23, 20, 32, 51].

## 8 Conclusion and future work

In this work, we extended randomized smoothing as a defense against additive norm-bounded adversarial attacks to the case of classification in the embedding space that is used in few-shot learning scenarios. We performed an analysis of Lipschitz continuity of smoothed normalized embeddings and derived a robustness certificate against $l_2-$attacks. Our theoretical findings are supported experimentally on several datasets. There are several directions for future work: our approach can possibly be extended to other types of attacks, such as semantic transformations; also, it is important to reduce the computational complexity of the certification procedure.

# 9   Acknowledgements

The work was supported by Ministry of Science and Higher Education grant No. 075-10-2021-068.

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
