# OpenReview forum: "Smoothed Embeddings for Certified Few-Shot Learning"
_NeurIPS.cc/2022/Conference — NeurIPS 2022 Accept_

### Official Review · Reviewer_d5wK · 2022-07-03

**Rating:** 7
**Confidence:** 4
**Soundness:** 2 fair
**Presentation:** 3 good
**Contribution:** 3 good

**Summary:**

The paper modifies the randomized smoothing robustness approach to a setting where a DNN computes an embedding and the classification is done via nearest neighbor approach to a class prototype. The paper derives a robustness guaranteed bound, and a way to estimate it with high probability.

**Questions:**

The main issue for me is the lack of any comparison. Secondary is fixing the confidence intervals to take into account that there are multiple estimations

It would also be beneficial to estimate the empirical robustness and see how tight the bounds are. I think this will improve the paper but isn't requires as the previous remark.

**Limitations:**

While the authors have a section on limitations, I don't think the discussion was on the real limitations of paper. The main limitation that I see is runtime and it should be investigated more thoroughly.

**Strengths And Weaknesses:**

Main strengths:
- The paper tackles the significant problem of robust learning in the few-shot case.
- While the paper builds on Cohen et al, it uses the properties on this specific case to get a simple bound. The contribution is somewhat novel.
- Quality of writing is mostly good

Main weaknesses:
- The paper does not compare his results on others, e.g. Kumar & Goldstein, so it is hard to know if the paper has any empirical significance. The theoretical contribution is not significant enough to carry this paper by itself.
- The confidence bound uses the Hoeffding inequality, but you use it to bound several estimations. In that case you need the union bound (or something better) to get a high-probability bound on all distances jointly.
- One main limitation is runtime as it can can reach 5 seconds for a single predictions. As the runtime is dynamic, it would be useful to show a breakdown of runtimes on the datasets.

Other remarks:
- You use $n$ with two different meanings which is confusing (the dimension $\mathbb{R}^n$ and the number of MC samples in eq. 10). Should change to avoid confusion.
- The algorithms written aren't accurate. You use set notation but it isn't correct the way you write it down.
- First paragraph isn't well-written and I would recommend editing it.
- Line 19 "This shows how easy..." needs rewriting
- Line 166 use $g_1,g_2$ without defining them.

---

> ### Author Response · Authors · 2022-08-01
> **Response to reviewer d5wK**
>
> We are grateful for your valuable feedback! In our response, we answer your questions and comment on your remarks.
>
> $\textbf{A1.}$ Indeed, the comparison with existing methods should be done to properly assess our approach. Thank you for mentioning the work [1] of Kumar and Goldstein that is perhaps the closest one to ours. We have performed additional experiments comparing our approach to the one from [1]. TLDR: our approach seems to be more efficient both computationally and in terms of certified accuracy. Details are below.
>
> In [1], authors developed Center Smoothing for certification against norm-bounded additive perturbations. This approach allows to certify models that map input object to metric spaces (such as image embeddings). Although the authors do not solve certified classification problem, their method can be adapted to it. However, the certification procedure in Center Smoothing only allows to bound the deviation $\varepsilon_{out}$ of model's output that its input is perturbed by the noise of the magnitude bounded from above by $\varepsilon_{in}$, what is sort of inverse problem to the one we solve -- given the magnitude of perturbation in the output space, produce a certificate for the input perturbation.
>
> To compare these two techniques, we did the following. Given the model to certify (smoothed prototypical network), we firstly sample a random subset of $N = 100$ images. Then, the model is certified (if possible) at each object $x$ from this subset by our method, so both $\gamma(x)$ and $\delta(x)$ are produced (here $\delta(x)$ is certified radius at point $x$ and $\gamma(x)$ is the maximum $\ell_2$-perturbation  of $\|g(x)-g(x+w)\|_2$ in the embedding space such that for $\|w\|_2 \le \delta(x)$,  $g(x+w)$ and $g(x)$ are assigned to the same class).
>
> Then, the model is certified (if possible) at $x$ with Center Smoothing and we compute the bound $\varepsilon_{out}(x)$ on  $\|g(x)-g(x+w)\|_2$.
>
> If $\varepsilon_{out}(x)$ is smaller than $\gamma(x)$, the model is considered certified at $x$, otherwise no. In other words, Center Smoothing tries to certify the model at points subject to perturbations that are certified by our method.
>
> In the Table below, we report the results of comparison, where SE is Smoothed Embeddings (our method) and CS is Center Smoothing. For all experiments, we fix the number of samples used in both certification procedures $n_{samples}=10^5$, the confidence levels for interval estimations $\alpha=10^{-4}$ and noise $\sigma=0.5$. Metrics are calculated as (note that the model may be certified only at points that are classified correctly):
>
> $$CA(SE) = \frac{\sum \mathbb{1}\\{x:\ !SE.abstain(x) \\; \\& \\;  y_{pred}=y_{true}\\}}{N}$$$$CA(CS) = \frac{\sum \mathbb{1}\\{x:\ \varepsilon_{out}(x) < \gamma(x)\\; \\& \\; y_{pred}=y_{true}\\}}{N}$$
>
> \begin{array}{|c|c|c|c|c|c|}\\hline Dateset & Shots & CA (SE) & CA (CS) & Time(SE), sec. & Time (CS), sec.\\\\\\hline\\hline CIFAR-FS & 1-shot & 39\\% & 23\\% & 0.507\pm0.157&53.725 \pm 0.231 \\\\\\hline&5-shot &16\\% &12\\% &0.834 \pm 0.182& 60.969\pm0.182\\\\\\hline Cub-200-2011 & 1-shot & 41\\%&38\\%&0.292\pm0.100&60.754 \pm 0.309\\\\\\hline& 5-shot & 17\\%&16\\%&0.348\pm0.084 &60.630\pm0.239\\\\\\hline\textit{mini}ImageNet &1-shot & 44\\% &11\\% & 0.329 \pm0.121& 61.470 \pm 0.217\\\\\\hline& 5-shot & 12\\% & 5\\%&1.279\pm0.322&61.226\pm 0.304\\\\\\hline\\hline\end{array}
>
> Note that for all experiments, our method marginally surpasses the one from [1]. More than that, the certification by SE is notably faster than the procedure from  CS. Perhaps the reason is  that the procedure from CS computes approximated minimum enclosing ball for each certified sample.
>
> $\textbf{Additional experiments.} $ Note that we have conducted additional experiments assessing the influence of the data augmentation during training and evaluated our approach against random and adversarial attack. Details are in the Appendix B.2.
> \\
>
> $\textbf{A2.}$ Please note that we use the Hoeffding inequality for the mean of observations (see [2], (2.6)). Please also note that for all the distances interval estimations are computed assuming that corresponding random variables (unbiased estimates of dot product of vectors) are independent. After interval estimations are computed, two closest prototypes are determined as described in Algorithm 1.
> \\
>
> Once again, thank you for careful reading. We have fixed the  notation in algorithms, the dimension of the input space (to avoid the confusion with the number of samples), the typo on the line $166$ (should be $g(x)$ instead of $g_1(x)$ and $g_2(x)$) and revisited the introduction. The text of the paper is updated.
> \\
>
> [1] Aounon Kumar and Tom Goldstein. "Center smoothing: Certified robustness for networks with structured outputs." NeurIPS,  2021.
>
> [2]. Hoeffding, Wassily. "Probability inequalities for sums of bounded random variables." The collected works of Wassily Hoeffding. Springer, New York, NY, 1994. 409-426.

---

> > ### Comment · Reviewer_d5wK · 2022-08-03
> > **Response to response**
> >
> > Thank you for the response and I did raise my score given your comparison to [1].
> >
> > Regarding the Hoeffdingf inequality - You have one bound per class (eq. 10). If you want to guarantee robustness then all bounds over all classes must hold uniformly. This means that you need the union bound (or an alternative) to guarantee robustness. This shouldn't be a drastic change, as it should add a log of the number of classes factor but should be fixed.

---

> > > ### Author Response · Authors · 2022-08-03
> > > **Confidence intervals: discussion.**
> > >
> > > Thank you for your clarification and for reconsidering your score!
> > >
> > > Speaking about Hoeffding inequality: as far as we understood, you are interested in the confidence level $\alpha^*$ for the uniform bound.
> > >
> > > The precise value of that probability (the probability of at least one bound not to hold)  is
> > > $p = 1 - (1-\alpha)^K$  where $\alpha$ is the confidence level for one bound and $K$ is the number of classes; this value has an natural upper bound $\hat{p} = K \alpha,$ that may be obtained either with Taylor series or by considering the worst situation -- when each bound does not hold independently.
> > >
> > > So $\alpha^* = K \alpha,$ where we have the factor of the number of classes, not log of it. Could you please elaborate on this multiplicative factor?

---

> > > > ### Comment · Reviewer_d5wK · 2022-08-04
> > > > **Confidence intervals: discussion II**
> > > >
> > > > First, taking the natural upper bound $K\alpha$ is the uniform bound.
> > > >
> > > > Second, the distince from the mean, i.e. the t in the Hoeffding inequality, is in exponent and the K is outside. So if we try to find the t such that $\exp\left(-\frac{2t^2n^2}{\sum(b_i-a_i(^2}\right)=K\alpha$ then the dependence of t on $K$ will be logarithmic.
> > > >
> > > > Also, I noticed that you use $\alpha$ twice in two separate meanings: Once from the probability of not being in the confidence interval and once for the lower bound on this interval. Please change it as it is confusing.

---

> > > > > ### Author Response · Authors · 2022-08-04
> > > > > **Confidence intervals: discussion III**
> > > > >
> > > > > Thank you for clarifying that it is the width of each confidence interval what has logarithmic dependence on $K.$
> > > > >
> > > > > To take into account the probability $K\alpha$ of at least one bound not to hold there is no need to increase the width of each interval, since $2\exp\left( - \frac{2t^2n^2}{\sum (b_i-a_i)^2}\right) = \alpha$ for single interval leads to  $K \alpha$ as the resulting failure probability.
> > > > >
> > > > > Still there is need to fix error probability of Algorithms in Sec. 6.1 for the given confidence level $\alpha$ for each interval.
> > > > >
> > > > > We have updated that section (to be consistent with algorithms description and graphs of experiments) and changed the notation of bounds of confidence intervals from $(\alpha, \beta)$ to $(l, u).$
> > > > >
> > > > >
> > > > > The text of the paper is updated.

---

> > > > > > ### Comment · Reviewer_d5wK · 2022-08-04
> > > > > > **Confidence intervals: discussion IV**
> > > > > >
> > > > > > Thanks you. After the corrections, how are your results compared to CS?

---

> > > > > > > ### Author Response · Authors · 2022-08-05
> > > > > > > **Confidence intervals: updated comparison to CS**
> > > > > > >
> > > > > > > To have the same error probability, we increased the confidence in CS procedure roughly in $K+1$ times.
> > > > > > >
> > > > > > >
> > > > > > > Updated results are in the table.
> > > > > > >
> > > > > > > \begin{array}{|c|c|c|c|c|c|}\\hline Dateset & Shots & CA (\textit{SE}) & CA (\textit{CS}) & Time(\textit{SE}), sec. & Time (\textit{CS}), sec.\\\\\\hline\\hline CIFAR-FS & 1-shot & 39\\% & 27\\% & 0.507\pm0.157&10.678 \pm 0.022 \\\\\\hline&5-shot &16\\% &15\\% &0.834 \pm 0.182& 10.979\pm0.016\\\\\\hline Cub-200-2011 & 1-shot & 41\\%&38\\%&0.292\pm0.100&10.535 \pm 0.018\\\\\\hline& 5-shot & 17\\%&17\\%&0.348\pm0.084 &10.539\pm0.035\\\\\\hline\textit{mini}ImageNet &1-shot & 44\\% &18\\% & 0.329 \pm0.121& 10.601 \pm 0.022\\\\\\hline& 5-shot & 12\\% & 5\\%&1.279\pm0.322&10.541\pm 0.016\\\\\\hline\\hline\end{array}
> > > > > > >
> > > > > > >
> > > > > > > Although the gap between accuracies decreased in some experiments, our method still outperforms CS. The main difference with the previous runs are in the execution times of CS -- perhaps with higher error probability and, thus, more narrow intervals, CS takes notably less time.

---

### Official Review · Reviewer_EGnM · 2022-07-07

**Rating:** 6
**Confidence:** 3
**Soundness:** 3 good
**Presentation:** 2 fair
**Contribution:** 3 good

**Summary:**

Applying randomized smoothing to few shot tasks is challenging since it heavily exploits the fact that classifiers are restricted to map an input to a fixed number of class probabilities. Previous work rely on improvement of empirical robustness or probabilistic guarantees of robustness. There is yet no theoretical guarantee on worst-case model behavior. The authors generalize and theoretically justify the idea of randomized smoothing to few-shot learning. Provable certification needs to be obtained not in the space of output class probabilities, but in the space of descriptive embeddings. Specifically, the authors generalize the analysis of Lipschitz-continuity to the case of vector functions and provide robustness guarantees for classification performed in the space of prototypical embeddings. This involves some adjustments to ensure the estimate of the Lipschitz constant is not simply an estimate of the norm of the multivariate Gaussian.

To provide certification, the authors define the adversarial embedding risk, which is the minimum L2 distortion in the embedding space required to change the prediction of $g$ (distance of the embedding $g(x)$ to the nearest decision boundary between nearest two class prototypes). In practice, the authors compute an estimate of the distance to the closest decision boundary from $g(x)$ as two-sided confidence intervals for each class prototype nearest to the embedding (and in practice the distance is estimated by computing confidence intervals for the dot product of those prototypes). It can be shown from Theorem 1 that the lower confidence bound gives the certified radius for the embedded sample. The authors provide an analysis of the failure rate for the practical implementation, as well showing that the approach achieves a reasonable certified accuracy and abstain rate.

**Questions:**

- How can the approach be generalized for other few-shot learning schemes?
- As previously stated, the proposed technique relies on prototypical learning embeddings (i.e., Theorem 1). In order to obtain a confidence interval for the distance between prototypes, the authors propose to compute confidence intervals for the dot product of vectors. Does this impose additional constraints on the posterior distribution of the embedding function? In practice the embedding function's posterior may not be well behaved, e.g,. is not centered at zero (all directions point towards same part of latent space) or has an elongated distribution (i.e., class centers are not decorrelated). Can the authors clarify these constraints?

**Limitations:**

- The authors have presented a first look at the certified robustness of few-shot learning models, although the treatment is limited to the context of prototypical learning. Some discussion of adaptation to other few-shot learning schemes would be helpful.
- The results should be fleshed out to compare against baseline natural model (no certified radius) and the certified model without data augmentation.

**Strengths And Weaknesses:**

Strengths:
-  The problem of performing certified robustness in the few-shot learning setting is presented in an intuitive way, primarily motivated by the difficulty of providing certified radius for samples embedded in a metric space, rather than fixed class probabilities. In this sense the problem the authors tackle is original and would be useful for the broader community.
- The authors provide an analysis of the Lipschitz-continuity of vector functions, which enables the lifting of previous results to the novel context of few-shot embedding and offers a significant contribution.
- A variety of experimental settings are studied, including CUB200, CIFAR-FS, and miniImagenet data. The results are structured similar to previous work and study the certified accuracy with respect to certification parameters, abstain rate, and runtime of the inference algorithm.


Weaknesses:
- The certified accuracy is quite low and seems to decrease linearly with $\epsilon$ despite the data augmentation. It is difficult to visualize the presented certified accuracy without the baseline comparisons (e.g., standard uncertified model with $\epsilon$ random noise added, and certified model without data augmentation).
- Unlike the previous techniques for certified robustness, the proposed method places some constraint on the underlying model, e.g., the assumption that the model performs prototypical learning. The authors do not discuss possible steps to generalize the proposed method to other few-shot techniques, e.g., model-agnostic meta-learning (MAML) or matching networks.
- Some issues with the organization prevent the reader from fully appreciating the content. For example, the proof for Theorem 1 is labeled as Theorem 4 in the Appendix, likewise the proof for Theorem 2 is labeled as Theorem 5. A full text revision would be necessary to fix some content issues, such as streamlining the introduction and Section 4 to improve clarity.

---

> ### Author Response · Authors · 2022-08-01
> **Response to reviewer EGnM**
>
> Thank you for you valuable feedback! In our response, we firstly answer you questions and then comment on weaknesses and limitations you mentioned.
>
> $\textbf{A1.}$ Among the approaches presented in [1], RS is applicable for matching networks and to some extent to MAML networks. In case of matching networks,  query sample is labeled as weighted cosine distance to support samples, there is no principal difference with the metric-based classification in our setting. For MAML, the smoothing may be applied to the embedding function as well. Regarding our method, it may be applicable for the matching networks (since it is easy to transfer guarantees from $l_2$ distance to cosine distance in case of normalized embedding), but for the MAML additional theoretical derivations are required.
> \\\\
>
>
> $\textbf{A2.}$ Please note that our approach does not impose significant constraints on the behavior of the embedding function. However, it's worth mentioning that we both (i) force the base model to map input points to the unit sphere and (ii) assume that the estimates of the squared distances between smoothed embedding and all the prototypes are independent random variables; our method also does not imply the posterior of embedding function to be centered at zero.
> \\\\
>
>
> Also we have performed additional experiments assessing our approach against random attacks, against adversarial attack and evaluate the effect of the data augmentation during the baseline model training. We have included results of experiments in the Appendix B, Subsection B.2. Below are the settings of experiments and conclusions.
>
>
> For each dataset, we have sampled a random subset $S$ of $500$ images and used it as a test set.
>
> The random attacks were conducted on the plain baseline model $f.$ In this experiment, the input $x$ of the model were perturbed by a random noise $\delta$ of the fixed magnitude. In this case, empirical robust accuracy is computed as follows:
>
> $$CA(S, \varepsilon) = \frac{|(x,y) \in S: h(f(x))=h(f(x+\delta))=y|}{|S|}, \\  \|\delta\|_2=\varepsilon$$
>
> Here and below, $h(\cdot)$ corresponds to the classification rule from Section 5.3.
>
> For the adversarial attack, the smoothed model $g$ was attacked at input point $x$ with the FGSM attack [2]. Namely, given an approximation of the smoothed model in the form $\hat{g}(x) = \frac{1}{n}\sum f(x+\varepsilon),$ the additive perturbation is computed as follows:
>
> $$\delta(x) = sign\left(\frac{1}{n} \sum \nabla_x f(x+\varepsilon)\right), $$
>
> so the additive perturbation corresponds to the projection of the mean gradient. In this case, the empirical robust accuracy looks as follows:
>
> $$CA(S, \varepsilon) = \frac{|(x,y) \in S: h(g(x))=h(g(x+\delta)) =y|}{|S|}, \  \|\delta\|_2=\varepsilon$$
>
> Also we have evaluated the effect of the data augmentation during the training of the baseline model. For all datasets for both $1-$shot and $5-$shot settings, we have trained new baseline models without data augmentation and evaluated associated smoothed models.
>
>
> The figures with the quantitative results are in appendix B. For all the smoothed models, we fixed the  confidence level $\alpha = 10^{-4}$ and number of samples $n=1000$. The variance $\sigma$ of additive noise for Cub-200-2011 and CIFAR-FS datasets is set  $\sigma=1.0,$ for $\textit{mini}$ImageNet dataset $\sigma=0.5.$
>
>
>
> From the pictures, we made several observations. Firstly, the augmentation of the baseline model with an additive noise increases empirical robustness of the smoothed model, so it is natural to augment the data to have both better robustness and larger accuracy when there is no attack. Secondly, even the baseline models could not be attacked with the random additive perturbation. Although the fact that the probability of randomly generated additive noise to be adversarial is low, it is interesting to show that it holds for the prototypical networks too. Finally, we observe that even  adversarial attack is not effective against smoothed models.
>
> Note that our theoretical guarantees are provided for the worst-case behavior of the smoothed models, in practice smoothed encoders may have very strong empirical robustness.
> \\
>
> [1] Triantafillou, E., Zhu, T., Dumoulin, V., Lamblin, P., Evci, U., Xu, K., ... \& Larochelle, H. (2019). Meta-dataset: A dataset of datasets for learning to learn from few examples. arXiv preprint arXiv:1903.03096.
>
> [2] Goodfellow, I. J., Shlens, J., \& Szegedy, C. (2014). Explaining and harnessing adversarial examples. arXiv preprint arXiv:1412.6572.
> \\
>
> Thank you for careful reading! We have revisited the text of the paper and fixed some issues both with notation and indexing. The text of the paper was updated.

---

> > ### Comment · Reviewer_EGnM · 2022-08-04
> > **Update and FGSM usage**
> >
> > A.1. Thanks to the authors for clarifying the relationship of the proposed technique to matching networks and MAML. Unless I missed it in the current draft, it would be helpful to include such a discussion in the main text.
> >
> > A.2. The new results are helpful to see the overall performance. Regarding the attack comparison, can the authors motivate the use of FGSM? Carlini et al. [1] have argued (page 12 of their paper) not to use FGSM exclusively in performance comparison, mainly since it was designed for speed rather than depth.
> >
> > [1] N. Carlini et al., “On Evaluating Adversarial Robustness,” arXiv:1902.06705 [cs, stat], Feb. 2019, http://arxiv.org/abs/1902.06705

---

> > > ### Author Response · Authors · 2022-08-05
> > > **Update and FGSM usage: response**
> > >
> > > 1) We have added the relationship of the technique to matching networks and MAML to the main text of the paper.
> > >
> > > 2) The choice of FGSM as adversarial attack is mainly motivated by the limited time for the additional experiments and the computational  complexity of evaluating the gradient of the smoothed model. Still, indeed the FGSM is by far not the strongest white-box attack, thank you for indicating that.

---

> > > > ### Comment · Reviewer_EGnM · 2022-08-07
> > > > **Attack baselines**
> > > >
> > > > Thanks for the response. I am more comfortable increasing the score based on the new text and evaluation. Still, the authors should compare against stronger attacks in the next revision to flesh out the evaluation, in line with the recommendations of Carlini et al. [1].
> > > >
> > > >
> > > > [1] N. Carlini et al., “On Evaluating Adversarial Robustness,” arXiv:1902.06705 [cs, stat], Feb. 2019, http://arxiv.org/abs/1902.06705

---

> > > > > ### Author Response · Authors · 2022-08-09
> > > > > **Attack baselines: response**
> > > > >
> > > > > Thank you for your response and for your suggestion. We have conducted additional experiments where we compare FGSM attack and its multi-step version, PGD attack (see Appendix B.2.1). Due to the limited time, we ran experiments for 1-shot settings only.
> > > > >
> > > > > In all experiments, we run PGD attack for $s=20$ iterations. Attack norm on each iteration is $s$ times smaller than the one in FGSM experiment; it is done to compare methods of attacks on corresponding magnitudes.
> > > > > For all the smoothed models, we fixed the
> > > > >  confidence level $\alpha=10^{-4}$ and number of samples $n = 1000.$ The variance $\sigma$ of additive noise for
> > > > > Cub-200-2011 and CIFAR-FS datasets is set $\sigma = 1.0$, for $\textit{mini}$ImageNet dataset $\sigma = 0.5.$
> > > > >
> > > > > It is notable that multi-step attack is more effective against smoothed model, especially on larger norms of perturbations. Still, the model performs well even against PGD attack.

---

### Official Review · Reviewer_1EMm · 2022-07-12

**Rating:** 7
**Confidence:** 4
**Soundness:** 4 excellent
**Presentation:** 4 excellent
**Contribution:** 4 excellent

**Summary:**

This paper proposes a certified robustness method for few-shot classification based on randomized smoothing. The paper focuses on prototypical classifiers where the classification is done based on the distance to the closest class prototype and class prototypes are the average of the features of class samples. Section 3 proposes two Theorems that guarantee the minimum perturbation size to change the label for a prototypical classifier to be proportional to the relative distance between the smoothed predicted features and two class prototypes. Section 4 proposes the estimation of the smooth prediction using finite samples and discusses a method based on confidence intervals to adaptively increase the number of samples until an acceptable confidence level is reached. Section 5 evaluates the effectiveness of the method on various few-shot learning benchmarks and shows improved certified accuracy for non-zero perturbation size.


**Questions:**

- Complementary to Tables 2-3, can you include the distribution of the smallest sample size at which the confidence level is reached on each test dataset?

**Strengths And Weaknesses:**

Strengths:
- Theoretical results and empirical evaluations successfully demonstrate the robustness of the method and provide a practical algorithm for robust few-shot learning.
- The paper is easy to read.
- To the extent that I know, the idea is novel for certified robustness in few-shot learning.
- To the extent that I checked, the theoretical results are correct and the algorithm follows the theory.

Weaknesses:
- It is important to verify the certificates by empirically attacking a model within the certified radius. It would be interesting to see if the certificate radius is tight by attacking the model with slightly larger perturbations.
- Related work is missing other methods common in few-shot learning and a discussion on whether randomized smoothing could be extended beyond prototypical networks. For example see methods evaluated in [1].

[1] Triantafillou, E., Zhu, T., Dumoulin, V., Lamblin, P., Evci, U., Xu, K., ... & Larochelle, H. (2019). Meta-dataset: A dataset of datasets for learning to learn from few examples. arXiv preprint arXiv:1903.03096.

---

> ### Author Response · Authors · 2022-08-01
> **Response to reviewer 1EMm**
>
> Thank you for your valuable feedback! In our response, we answer your question and comment on the evaluating our method against random attacks, adversarial attacks and assess the effect of data augmentation.
>
>
> $\textbf{A1.}$ We have added the distribution of  the smallest sample size at which the confidence level is reached to the section Additional experiments in the appendix of the paper (Subsection B.1). All the experiments were conducted with the following parameters: the variance of additive noise $\sigma=1.0,$ maximum number of samples $n=100000$. Note that for all the experiments and for all values $\alpha$ of interest, the required number of samples is less than $10000$ for most of the input images.
> \\
>
> Also we have performed additional experiments assessing our approach against random attacks, against adversarial attack and evaluate the effect of the data augmentation during the baseline model training. We have included results of experiments in the Appendix B, Subsection B.2. Below are the settings of experiments and conclusions.
>
>
> For each dataset, we have sampled a random subset $S$ of $500$ images and used it as a test set.
>
> The random attacks were conducted on the plain baseline model $f.$ In this experiment, the input $x$ of the model were perturbed by a random noise $\delta$ of the fixed magnitude. In this case, empirical robust accuracy is computed as follows:
>
> $$
> CA(S, \varepsilon) = \frac{|(x,y) \in S: h(f(x))=h(f(x+\delta))=y|}{|S|}, \  \|\delta\|_2=\varepsilon
> $$
>
> Here and below, $h(\cdot)$ corresponds to the classification rule from Section 5.3.
>
> For the adversarial attack, the smoothed model $g$ was attacked at input point $x$ with the FGSM attack [2]. Namely, given an approximation of the smoothed model in the form $\hat{g}(x) = \frac{1}{n}\sum f(x+\varepsilon),$ the additive perturbation is computed as follows:
>
> $$
>     \delta(x) = sign\left(\frac{1}{n} \sum \nabla_x f(x+\varepsilon)\right),
> $$
>
> so the additive perturbation corresponds to the projection of the mean gradient. In this case, the empirical robust accuracy looks as follows:
>
> $$
> CA(S, \varepsilon) = \frac{|(x,y) \in S: h(g(x))=h(g(x+\delta)) =y|}{|S|}, \  \|\delta\|_2=\varepsilon
> $$
>
>
> Also we have evaluated the effect of the data augmentation during the training of the baseline model. For all datasets for both $1-$shot and $5-$shot settings, we have trained new baseline models without data augmentation and evaluated associated smoothed models.
>
>
> The figures with the quantitative results are in appendix B. For all the smoothed models, we fixed the  confidence level $\alpha = 10^{-4}$ and number of samples $n=1000$. The variance $\sigma$ of additive noise for Cub-200-2011 and CIFAR-FS datasets is set  $\sigma=1.0,$ for $\textit{mini}$ImageNet dataset $\sigma=0.5.$
>
>
>
> From the pictures, we made several observations. Firstly, the augmentation of the baseline model with an additive noise increases the empirical robustness of the smoothed model, so it is natural to augment the data to have both better robustness and larger accuracy when there is no attack. Secondly, even the baseline models could not be attacked with the random additive perturbation. Although the fact that the probability of randomly generated additive noise to be adversarial is low, it is interesting to show that it holds for the prototypical networks too. Finally, we observe that even  adversarial attack is not effective against smoothed models.
>
> Note that our theoretical guarantees are provided for the worst-case behavior of the smoothed models, in practice smoothed encoders may have very strong empirical robustness.
> \\
>
> Thank you for indicating the importance to discuss whether RS is applicable for other few-shot techniques. Among the approaches presented in [1], RS is applicable for matching networks and to some extent to MAML networks. In case of matching networks,  query sample is labeled as weighted cosine distance to support samples, there is no principal difference with the metric-based classification in our setting. For MAML, the smoothing may be applied to the embedding function as well. Regarding our method, it may be applicable for the matching networks (since it is easy to transfer guarantees from $l_2$ distance to cosine distance in case of normalized embedding), but for the MAML additional theoretical derivations are required.
> \\
>
>
> [1] Triantafillou, E., Zhu, T., Dumoulin, V., Lamblin, P., Evci, U., Xu, K., ... \& Larochelle, H. (2019). Meta-dataset: A dataset of datasets for learning to learn from few examples. arXiv preprint arXiv:1903.03096.
>
> [2] Goodfellow, I. J., Shlens, J., \& Szegedy, C. (2014). Explaining and harnessing adversarial examples. arXiv preprint arXiv:1412.6572.

---

> > ### Comment · Reviewer_1EMm · 2022-08-07
> > **Thank you for your response**
> >
> > Thank you for the additional experiments. I further support the suggestion from reviewer EGnM to include evaluations against attacks stronger than FGSM. Overall, I retain my score and recommendation for acceptance.

---

> > > ### Author Response · Authors · 2022-08-09
> > > **Response to reviewer 1EMm II**
> > >
> > > Thank you for your response. We have conducted additional experiments where we compare FGSM attack and PGD attack (see Appendix B.2.1).
> > >
> > > In all experiments, we run PGD attack for $s=20$ iterations. Attack norm on each iteration is $s$ times smaller than the one in FGSM experiment; it is done to compare methods of attacks on corresponding magnitudes.
> > > For all the smoothed models, we fixed the
> > >  confidence level $\alpha=10^{-4}$ and number of samples $n = 1000.$ The variance $\sigma$ of additive noise for
> > > Cub-200-2011 and CIFAR-FS datasets is set $\sigma = 1.0$, for $\textit{mini}$ImageNet dataset $\sigma = 0.5.$
> > >
> > > It is notable that multi-step attack is more effective against smoothed model, especially on larger norms of perturbations. Still, the model performs well even against PGD attack.

---

> ### Author Response · Authors · 2022-08-07
> **Response to reviewer 1EMm II**
>
> Dear reviewer 1EMm,
>
> Did you have a chance to take a look at our response, does it clarify your questions?

---

### Official Review · Reviewer_JtzZ · 2022-07-13

**Rating:** 5
**Confidence:** 2
**Soundness:** 3 good
**Presentation:** 3 good
**Contribution:** 3 good

**Summary:**

In this work, the authors extend the idea of randomized smoothing (used to tackle adversarial perturbations) to the paradigm of metric-learning based few-shot learning algorithms (e.g. Prototypical Networks). In particular, they propose an algorithm regarding how randomized smoothing can be used with vector-valued functions (e.g. the ones that compute the embeddings) which are used in Prototypical Networks to compute the distances of each query samples against the prototypes. Authors provide theoretical justification regarding their proposed algorithm. Empirical validation is also performed using multiple datasets.

**Questions:**

* For theorem 1, one of the assumptions is that $\lVert f(x) \rVert_2 = 1$ - is this assumption true because the embeddings are normalized to unit norm before computing the L2 distance?
* For Mini-ImageNet, the experiments section mentions using 80 classes for training whereas in the official benchmark, there are 64 classes for training and 16 for validation. Was the model trained on the combined train + val set? I am asking this because for CIFAR-FS, it seems like the model was trained only on the 64 train classes (which is the common practice in few-shot learning literature).

**Ethics Review Area:**

["I don’t know"]

**Limitations:**

Same as my comments on weaknesses above. I will consider changing my vote if authors can provide comparisons against existing methods from the literature or convince me why this method can not compared against any other method.

**Strengths And Weaknesses:**

Strengths
------------
* The contribution in terms of the proposed algorithm to extend randomized smoothing to metric-learning based algorithms seems novel. Authors also provided theoretical justification to show that the proposed method can be provably used as a defense against adversarial attacks.
* The proposed algorithm is clearly explained, intuitively makes sense and also computationally cheap - thereby having a higher potential of being adopted in the community.
*  Experiments have been performed on multiple datasets from different domains and the method works consistently across these tasks.

Weaknesses
---------------
* The proposed algorithm is not compared against any other method from the literature. I understand it's a new problem area (randomized smoothing for vector-valued function) but I wonder if any other adversarial robustness algorithm(s) can be used to understand the relative performance gain obtained by this method.
* Although the paper mentions working in a few-shot learning setup, I do not see a clear reason why the method needs to be restricted to few-shot. The approach should work with most metric-learning algorithms and should be evaluated on those to understand how it works with those methods.

---

> ### Author Response · Authors · 2022-08-01
> **Response to reviewer JtzZ**
>
> Thank you for your valuable feedback! We start by answering your questions and then proceed to comparison to existing work.
>
> $\textbf{Q1.}$For theorem 1, one of the assumptions is that $\|f(x)\|_2=1$ is this assumption true because the embeddings are normalized to unit norm before computing the L2 distance?
>
> $\textbf{A1.}$ Yes, the output of the base model $f(\cdot)$ is normalized to the unit norm.
>
> $\textbf{Q2.}$ For Mini-ImageNet, the experiments section mentions using 80 classes for training whereas in the official benchmark, there are 64 classes for training and 16 for validation. Was the model trained on the combined train + val set? I am asking this because for CIFAR-FS, it seems like the model was trained only on the 64 train classes.
>
> $\textbf{A2.}$ Thank you for careful reading! Yes, Mini-ImageNet contains $64$ classes for training. It's a typo, for all the experiments, the models were indeed trained solely on training sets.
>
> $\textbf{Baseline comparison:}$
>
> Indeed, the problem we are solving is to some extent barely studied in the literature. For this reason, we compare our approach to perhaps the closest work of Kumar et al [1]. TLDR: our approach seems to be more efficient both computationally and in terms of certified accuracy. Details are below.
>
> In [1], authors developed $\textit{Center Smoothing}$ for certification against norm-bounded additive perturbations. This approach allows to certify models that map input object to metric spaces (such as image embeddings). Although the authors do not focus at certified classification problem, their method can be adapted to it. However, the certification procedure in  Center Smoothing only allows to bound the deviation $\varepsilon_{out}$ of model's output that its input is perturbed by the noise of the magnitude bounded from above by $\varepsilon_{in}$, what is sort of inverse problem to the one we are solving -- given the magnitude of perturbation in the output space, produce a certificate for the perturbation of input.
>
> To compare these two techniques, we did the following. Given the model to certify (smoothed prototypical network), we firstly sample a random subset of $N = 100$ images. Then, the model is certified (if possible) at each object $x$ from this subset by our method, so both $\gamma(x)$ and $\delta(x)$ are produced (here $\delta(x)$ is certified radius at point $x$ and $\gamma(x)$ is the maximum $\ell_2$-perturbation  of $\|g(x)-g(x+w)\|_2$ in the embedding space such that for $\|w\|_2 \le \delta(x)$,  $g(x+w)$ and $g(x)$ are assigned to the same class).
>
> Then, the model is certified (if possible) at $x$ with Center Smoothing and we compute the bound $\varepsilon_{out}(x)$ on  $\|g(x)-g(x+w)\|_2$.
>
> If $\varepsilon_{out}(x)$ is $\textit{smaller}$ than $\gamma(x)$, the model is considered certified at $x$, otherwise no. In other words, Center Smoothing tries to certify the model at points subject to perturbations that are certified by our method.
>
> In the Table below, we report the results of comparison, where $\textit{SE}$ is Smoothed Embeddings (our method) and $\textit{CS}$ is Center Smoothing. For all experiments, we fix the number of samples used in both certification procedures $n_{samples}=10^5$, the confidence levels for interval estimations $\alpha=10^{-4}$ and noise $\sigma=0.5$. Metrics are calculated as (note that the model may be certified only at points that are classified correctly):
>
> $$CA(\textit{SE}) = \frac{\sum \mathbb{1}\\{x:\ !SE.abstain(x) \\; \\& \\;  y_{pred}=y_{true}\\}}{N}$$$$CA(\textit{CS}) = \frac{\sum \mathbb{1}\\{x:\ \varepsilon_{out}(x) < \gamma(x)\\; \\& \\; y_{pred}=y_{true}\\}}{N}$$
>
> \begin{array}{|c|c|c|c|c|c|}\\hline Dateset & Shots & CA (\textit{SE}) & CA (\textit{CS}) & Time(\textit{SE}), sec. & Time (\textit{CS}), sec.\\\\\\hline\\hline CIFAR-FS & 1-shot & 39\\% & 23\\% & 0.507\pm0.157&53.725 \pm 0.231 \\\\\\hline&5-shot &16\\% &12\\% &0.834 \pm 0.182& 60.969\pm0.182\\\\\\hline Cub-200-2011 & 1-shot & 41\\%&38\\%&0.292\pm0.100&60.754 \pm 0.309\\\\\\hline& 5-shot & 17\\%&16\\%&0.348\pm0.084 &60.630\pm0.239\\\\\\hline\textit{mini}ImageNet &1-shot & 44\\% &11\\% & 0.329 \pm0.121& 61.470 \pm 0.217\\\\\\hline& 5-shot & 12\\% & 5\\%&1.279\pm0.322&61.226\pm 0.304\\\\\\hline\\hline\end{array}
>
> Note that for all experiments, our method marginally surpasses the one from [1]. More than that, the certification by $\textit{SE}$ is notably faster than the procedure from  $\textit{CS}$. Perhaps the reason is  that the procedure from $\textit{CS}$ computes approximated minimum enclosing ball for each certified sample.
>
> In conclusion, we want to outline that our approach indeed can be used with other metric-learning algorithms, but the evaluation will require significant additional experiments. However, we still believe that the few-shot setting we focus in this paper is important.
>
> [1] Aounon Kumar and Tom Goldstein. "Center smoothing: Certified robustness for networks with structured outputs." NeurIPS,  2021.

---

> ### Author Response · Authors · 2022-08-07
> **Response to reviewer JtzZ II**
>
> Dear reviewer JtzZ,
>
> Did you have a chance to take a look at our response, does it clarify your questions?

---

> > ### Comment · Reviewer_JtzZ · 2022-08-08
> > **Thanks for your rebuttal against my review**
> >
> > First of all, thanks for your rebuttal against my review comments and taking the time to run additional experiments to compare against a baseline. Based on that and your discussion with other reviewers, I'll increase my score by one point.

---

> > > ### Author Response · Authors · 2022-08-08
> > > **Response to reviewer JtzZ III**
> > >
> > > Thank you for your review and for reconsidering your score!

---

### Meta-Review · Area_Chair_1fC7 · 2022-08-30

**Recommendation:** Accept
**Confidence:** Certain

**Metareview:**

This paper proposes a certified robustness method for few-shot learning classification based on randomized smoothing. The reviewers found the theoretical results and empirical evaluations successful in demonstrating the robustness of the method, providing a practical algorithm for robust few-shot learning problem. There were some concerns about the lack of comparison against other methods from the literature. But the authors addressed the issue in the rebuttal by running some additional experiments. The reviewers suggested that the authors motivate the use of FGSM and evaluate their method against other attacks as well.

**Award:**

No

---

### Decision · Program_Chairs · 2022-09-14

Accept